# Telehealth Intervention: A Proposal for a Telemedicine Manual to Ascertain the Civil Disability Status in Italy

**DOI:** 10.3390/ijerph21030253

**Published:** 2024-02-22

**Authors:** Nicola Di Fazio, Giuseppe Delogu, Donato Morena, Eugenia Carfora, Dalila Tripi, Raffaella Rinaldi, Paola Frati, Vittorio Fineschi

**Affiliations:** Department of Anatomical, Histological, Forensic and Orthopaedic Sciences, Sapienza University of Rome, 00128 Rome, Italy; giuseppe.delogu@uniroma1.it (G.D.); donato.morena@uniroma1.it (D.M.); eugenia.carfora@uniroma1.it (E.C.); dalila.tripi@uniroma1.it (D.T.); raffa.rinaldi@uniroma1.it (R.R.); paola.frati@uniroma1.it (P.F.); vittorio.fineschi@uniroma1.it (V.F.)

**Keywords:** telemedicine, civil disability, social welfare, protocol, legal medicine, Skype

## Abstract

(1) This paper aims to propose a telematic method for assessing the state of disability by a social worker member of the Medico-Legal Commissions of INPS. (2) We have proceeded to delve into and compare the current methods with new strategies to enhance the experience linked to the assessment of the status of disability in terms of timing and accessibility, eliminating the need for a direct visit. (3) The proposed protocol can be applied in cases where patients cannot be physically moved, following a medical visit at home, and when the mere evaluation of documents is sufficient. In both cases, a remote session with the Commission is necessary to fill in the required information about the socio-environmental section. This protocol can be applied to different platforms such as Skype™ since it is free and widely used throughout the country. (4) It should be noted that telemedicine solutions cannot completely replace face-to-face interaction; however, in some limited cases, they can optimize the process and timing, avoiding the need to move or interact only by telephone.

## 1. Introduction

Telemedicine enables the provision of healthcare services at distance through the use of the latest-generation technologies. It is utilized in situations where healthcare professionals and patients are not in the same location or in a circumstance that hinders a visit from being conducted in person [1].

Technological innovations in medical care have resulted in the development of telehealth programs—defined as “the use of electronic information and telecommunication technologies to support and promote long-distance clinical healthcare, patient and professional health-related education, public health and health administration” (available online: https://www.hrsa.gov/telehealth, accessed on 16 February 2024)—in both rural and urban settings. The genesis of such systems was as a means of providing access to health resources for individuals residing in isolated areas who cannot reach hospitals or clinics. Subsequently, it found widespread use in war and military territories [2,3]. Additionally, it has become a resource in urban centers for citizens unable to access health facilities for various reasons. It also serves as a platform for off-site doctors to conduct patient consultations at a distance [4].

Urban telehealth systems, monitored in institutions like Mercy Health System (Philadelphia, PA, USA) and AtlantiCare Regional Medical Center (Atlantic City, NJ, USA), demonstrate the significant benefits of telemedicine. It serves as a preliminary patient analysis tool for treating various conditions, particularly chronic, medical, psychiatric, and neurological pathologies [5,6].

However, the launch of telemedicine programs necessitates the simultaneous development of new “Ad hoc protocols” and safeguards. These are vital to protect patient privacy, ensure the adequate preparation of professionals practicing in the medical profession, and educate patients on the use of new technological systems [7].

Telemedicine must adhere to all the rights and obligations associated with any medical procedure, including providing comprehensive information to the patient. Therefore, patients must be informed about the advisability of telemedicine, as well as the methods used and the procedures for storing and processing personal data [8]. This should be done in compliance with the regulations in force at the national and regional levels.

Telemedicine serves as valuable support for recording and monitoring data of chronic patients. Electronic health diaries have enhanced the quality of information, enabling better data management and the creation of a robust network of contacts, prioritizing consultations. This network is capable of facilitating timely interventions and rapid connections, linking different hospital centers and patients’ homes [9].

However, the widespread use of telemedicine/telemonitoring services raises new ethical concerns, particularly concerning the evolving relationship between patients and doctors [10]. It is essential to consider the socio-cultural characteristics of various patients to make information as comprehensive as possible. To address user concerns and strengthen their confidence, it is necessary to implement promotion and information programs. These programs should allow patients to easily engage with various telemedicine platforms, familiarizing themselves with these new methods and tools. Similarly, healthcare professionals must approach and become accustomed to new programs to create a smart and comfortable environment during telematic visits (available online: https://www.salute.gov.it/portale/ehealth/homeEHealth.jsp, accessed on 16 February 2024).

Therefore, telemedicine can contribute to

Improving the quality of healthcare;Improving the usability of treatments, diagnostic services, and remote medical consultancy;The constant monitoring of vital parameters to reduce the risk of complications in people at risk or suffering from chronic diseases.

Telemedicine, representing one of the main areas of application of online healthcare, offers highly significant potential, especially for

Increasing equity in access to social and healthcare services in remote areas, thanks to the decentralization and flexibility of the services offered, made possible through innovative forms of domiciliary care;Optimally redistributing human and technological resources between different facilities, covering the need for professional skills that are often lacking, and ensuring continuity of assistance in the territory;Offering valid support for emergency mobile services through the availability of teleconsultation services. This is achieved by reorganizing health services, possibly utilizing remote clinical resources, even located directly on board ambulances.

The main disadvantage of this method is surely represented by the impossibility of seeing the patient in person. In fact, telemedicine does not replace basic visits where it is necessary to implement semiological maneuvers useful for a correct classification of the patient and, therefore, diagnosis. For this reason, its objective is the remote monitoring of patients with chronic diseases and, consequently, basic follow-up.

The history of telemedicine, on the other hand, has its foundations in the distant past. The first examples date back to the late nineteenth century, with the implementation of telephone consultations aimed at reducing unnecessary medical examinations. By the late 1950s, the first telematic communication system was developed by the Nebraska Psychiatric Institute and Norfolk State Hospital for psychiatric consultations [2].

In Italy, however, the implementation of telematic services within the National Health Service (SSN) only occurred in December 2020 (available online: https://www.agendadigitale.eu/sanita/telemedicina-come-farla-in-italia-le-tecnologie-le-finalita-un-modello-possibile/, accessed on 16 February 2024). Two years after the Ministry of Health’s endorsement, the first guidelines for television, teleconsultancy, telemonitoring, and remote assistance were edited (available online: https://www.salute.gov.it/imgs/C_17_pubblicazioni_2129_allegato.pdf, accessed on 16 February 2024).

These directives became necessary due to the pandemic experience. The outpatient medico-legal activity of the Italian National Institute of Social Welfare (INPS) faced a significant slowdown during the health emergency linked to SARS-CoV-2, especially in the initial pandemic phase when direct visits were restricted after the closure of premises [11,12]. Following the resumption of in-person activity, health restrictions to protect users and operators added 15–20 days to the average time of concluding the health process for first-instance applications, which in 2019 took about 3 months.

To address this issue, the Medico-Legal General Coordination (CGML) of the INPS instructed the Commissions at the Operative Medico-Legal Units from the beginning of the pandemic to define, where possible, assessments on acts also through remote activities. Subsequently, an important piece of legislation, effective since 15 September 2020, recognized and regulated the activity of acts of the Public Medical Commissions responsible for the acceptance of civil and disability rights (art. 29-ter Law 11/09/20 n. 120).

Based on this premise, this paper aims to identify a framework of solutions—both organizational and technological, taking into account the requirements of personal data protection—that allows the Medico-Legal Commissions of INPS to engage with involved parties in the process of determining disability through telemedicine.

The assessment of disability within the INPS focuses on formulating judgments on handicap conditions within a serious ‘situation.’ In this context, it is essential to emphasize not only the evaluation of impairments but also the consideration of socio-relational and environmental handicaps. Consequently, the opinion expressed by the social worker—documented in a file related to ‘activities and participation’ and ‘environmental factors,’ integral components of the disability report—is crucial. This form aims to codify the functioning profile of an individual and identify whether the environment acts as a ‘barrier’ or a ‘facilitator’ for them. Therefore, except for specific cases, the assessment must be personalized and cannot overlook the detailed description of the ‘functioning profile’ of the subject, resulting from the interaction of their impairments and abilities (previously assessed by the medical component of the Commission) with the obstacles and supports provided by environmental factors.

According to the current scenario, the Interdisciplinary Commission comprises three doctors (art. 1, paragraph 2, of Law 295/90) (available online: https://www.gazzettaufficiale.it/eli/id/1990/10/20/090G0349/sg, accessed on 16 February 2024): a specialist in forensic medicine who assumes the role of president and two doctors, one of whom is primarily chosen among specialists in occupational medicine. The Commission is supplemented by a social worker and possibly also by an expert in the specific case to be evaluated in the case of handicap status verification.

The first step of the procedure is the submission of the assessment request by the patient, while a qualified external doctor independently provides the submission of the introductory medical certificate. If the patient is deemed “not transportable” according to the introductory medical certificate, a home visit can be scheduled; otherwise, INPS forwards the communication to the Local Health Authority (ASL), and the visit date is scheduled in chronological order based on the submission of applications.

After examining the introductory certificate and the documentation sent by the interested party, the Commission assesses whether a visit in person is necessary. Another possibility is that, if the medical documentation is deemed sufficient by the President of the Medical Commission, a visit is not scheduled, and an evaluation is carried out exclusively “on acts.” In any case, at the end of the procedure, the Medical Committee shall draft the examination report, including the final judgment.

When the patient’s evaluation specifically concerns the judgment of handicap and the relative degree of severity, it is necessary to further investigate the socio-economic compromise and the relative environmental disadvantage faced by the subject. This is accomplished by completing a form provided by the International Classification of Functioning, Disability, and Health (ICF) Checklist of the World Health Organization (WHO) (available online: https://www.who.int/standards/classifications/international-classification-of-functioning-disability-and-health, accessed on 16 February 2024). All of this is aimed at outlining the subject’s ‘operating profile,’ based on the limitations in activity, restrictions in participation, and environmental factors.

If the evaluation is solely based on documents or a home visit is conducted, the evaluation by the social worker, with the consequent drafting of the ‘Social Form,’ includes a telephone conversation with the person concerned and/or their family members and caregivers. However, telephone interaction with the interested party alone has obvious limitations, as it does not allow for an overall assessment of the environmental, logistic, and social context of the interested party, which is necessary for determining the handicap as a social disadvantage of the subject in his own relational and cultural context.

The next paragraph will introduce a procedure that involves conducting a telematic examination with a communication platform capable of allowing a more complete assessment of the social, environmental, and family context of the patient in question.

## 2. Materials and Methods

The present work originates from a proposal for a manual of telematic examinations for the assessment of the state of disability for the social worker member of the INPS Medico-Legal Commissions. The original protocol was written by a commission composed of several members, some of whom are listed as authors in this paper.

Therefore, through the examination of this document, we proceeded to

Deepen the current methods of assessment and analyze the criticisms;Highlight all possible fields of implementation of telehealth provisions;Conceptualize a protocol, including the proposal of specific telematic platforms;Compare the old strategies with the new proposal, analyzing technological and implementation resources, to improve—in terms of timing and accessibility to health personnel and the public user of services—the experience linked to the assessment of the status of disability, where no direct visit of the patient is necessary.

Everything has been structured in the form of a commentary, divided into an introductory section containing the definitions of the key concepts related to telemedicine, a results section oriented to the introduction of potential new methods and strategies in terms of telematic assessment of the state of disability, a discussion section in which reflections are prepared on the subject, and a conclusion oriented to the question about the need for further improvements to the provision of telesessions.

At the moment, the outlined protocol represents an open proposal to be reviewed by the INPS Advisory Board. If the project is approved, a pilot phase for the implementation of the small-scale protocol is envisaged. During this phase, it will be the responsibility of the operators involved in the visits to complete a specially prepared Excel file to collect fundamental data for analyzing the effectiveness of the procedure from a multidisciplinary perspective.

As this protocol is intended to become the standard in practice, the analysis of items will focus on constructing statistical indicators of satisfaction, organization, and effectiveness for prospective monitoring over time and across different geographical realities.

## 3. Results

### 3.1. Telemedicine Application Scenario

Currently, the protocol proposed below, pertaining to the interview and analyses conducted by the Commission for the assessment of the handicap and by the social worker, is applicable in the following scenarios:When the mere evaluation of the documents is sufficient, and therefore, a remote session with the Commission is necessary for compiling the socio-environmental file.In the case of a transportable patient, following the medical visit at home, the tele-session with the Commission will continue for the compilation of the socio-environmental file.

The proposed scenario can be applied to different telematic platforms that are commonly used and easily available. Specifically, the software recommended in the implementation phase within the illustrated document is Skype™ [13], as it is free and widely accessible throughout the country. Numerous studies and international scientific experiences have demonstrated the validity of this platform in healthcare settings, having been applied for various purposes with encouraging results: virtual rehabilitation interventions for language disorders, remote synchronous consultations in Ophthalmic Telesurgery, health status monitoring following different kinds of interventions, psychotherapy, virtual consultations in Plastic Surgery, as well as in several clinical settings [14,15,16,17,18,19,20,21,22]. The choice of the platform is therefore based on the evidence of its validity, cost, and widespread availability within the territory.

On the other hand, the simple requirements for being able to use the depicted platforms correspond to those of any other method that can be used in telemedicine: a computer equipped with a webcam or a smartphone and an adequate internet connection. Of course, if it becomes necessary to implement additional communication channels, it is possible to include additional programs such as Zoom© or Google Meet™ which are both compliant with the European Union’s General Data Protection Regulation (GDPR) (available online: https://support.google.com/a/answer/7582940?hl=en, accessed on 16 February 2024 and https://explore.zoom.us/en/gdpr/, accessed on 16 February 2024).

Ultimately, the advantages of the described provider are that it

Can be used in the Institute;Is free to download;Is available in Italian and is already widespread and well known, and in any case sufficiently intuitive and easy to use, even by patients who are not particularly familiar with computer tools;Allows interaction without the need for a dedicated (personal or corporate) telephone number shared with the subject being investigated;Presents, as part of the intended use, appropriate security levels, implementing all communications based on Advanced Encryption Standard (AES) algorithms with strong 256-bit encryption (available online: https://support.Skype.com/it/faq/FA31/Skype-usa-la-crittografia, accessed on 16 February 2024), thus avoiding the risk of eavesdropping.

The disadvantage linked to this method, unresolved at the moment, is the limitation of internet coverage within the national territory. However, the advancement of technologies and the expansion of the communication network will make it possible in the future to considerably reduce or eliminate this aspect.

In any case, the patient subjected to the assessment has the right to refuse the interview in telematic mode and to request an appointment in person at the reference center.

#### 3.1.1. Case of Assessment Completion Entirely on Records

According to the theory proposed in this document, the case study is intended to be conducted based on the sole review of remote health documentation, and it is considered not feasible only if such documentation cannot be transmitted to the requesting body or is deemed insufficient. In this way, the assessment can be performed in compliance with the following procedural steps:The Medico-Legal Commissions send a letter to the interested party by email or postal mail (both in the case of a first assessment and in the case of revision), requesting health documentation. This follows the procedures established by the Institute.If the interested party does not provide probative documentation necessary for an objective evaluation, they will be summoned for a direct visit.If, on the other hand, the interested party submits adequate documentation within the established terms, or if suitable documentation is available to the Commission (e.g., civil disability report from a recent home visit), the Commission may draft the disability report based on the proceedings in a special session.The social worker contacts the applicant by telephone to inform them of the possibility of proceeding with the collection of socio-environmental elements through a telematic examination. In the case of acceptance, the social worker checks with the interested party the availability of the necessary technical equipment (such as a simple smartphone), explains the operations, and sends, by mail, an information letter pursuant to the protection of personal data and a brief instruction manual on how to use the platform and access the service. The date for a preliminary contact to verify the correct functioning of the communication platform and the subsequent date of the actual telematic examination are also agreed upon.Before the telematic examination, the social worker plans the video call, simultaneously sending the applicant a message via the platform, to which the applicant will respond for confirmation.Once the electronic connection with the interested party is established, the Commission will proceed with the interview to acquire socio-environmental elements, which will be recorded by the social worker in the appropriate form already present in the civil disability procedure, as an integral attachment to the final report.At the end of the interview, the social worker asks the patient to express their level of satisfaction with the procedure used and, if necessary, report any problems encountered. These data, together with other indicators evaluated by the Commission, are recorded in a special anonymous form to examine the validity of the telemedicine protocol used. Alternatively, it is possible to send the subject a questionnaire to be completed anonymously through easy-to-use platforms (e.g., Google Forms).Immediately after the telematic examination, the Commission will collectively complete the final minutes in the usual manner.

#### 3.1.2. Case of Completion of Home Visits (Non-Transferable Person)

This case undoubtedly represents a fundamental aspect of the proposed telematic system, as it would provide greater flexibility and convenience in the execution of the assessment by the personnel involved, with clear advantages in terms of efficiency and effectiveness of the work of the Social Operator.

The two doctors in charge, on the day of the home visit, ascertain that the person visited is able to use remote services, deliver the required documentation, and inform the interested party that they will be contacted by the social worker to proceed with the planning of the telematic examination with the Commission for the collection of socio-environmental elements.

Subsequent activities are carried out as described from point “D” onwards in the assessment of acts.

### 3.2. Protection of Personal Data

Regarding EU Regulation 2016/679 on the protection of personal data, the most relevant aspects are highlighted below (available online: https://eur-lex.europa.eu/legal-content/EN/ALL/?uri=celex%3A32016R0679, accessed on 16 February 2024):The National Institute of Social Security is responsible for processing personal data and data related to the health of the interested party.Before telematic contact, the subject undergoing assessment is informed of all the contents provided for in Article 13 of the Regulation.A letter describing the specific operational context within which the visit takes place and its implications in terms of personal treatment is sent to the subject.Further consent from the subject for the verification of personal data is not necessary, as the collection of such data takes place in the same way and for the same purpose as the in-person visit.Since the process does not collect additional data or involve processing other than the usual face-to-face visit, there is no high risk to the rights and freedoms of natural persons. Additionally, it should be noted that telematic platforms such as Skype™ have additional security mechanisms and organizational procedures suitable for collecting such data.An assessment has been made of the confidentiality and security criteria of the Skype™ platform, which are considered adequate as they are based on the encryption of communications.

## 4. Discussion

It should be considered that the proposed procedure is not exempt from currently insurmountable limitations. The medico-legal assessment, when not feasible through the sole analysis of documents, necessarily requires a direct visit from the interested party and is therefore not implementable through a telemedicine protocol. However, a different scenario arises concerning the interview and analysis conducted by the Disability Assessment Commission, specifically by the social worker, for the purpose of evaluating the context and subsequently compiling the socio-environmental form of the report. Since a personal clinical examination is not required, this activity can be implemented through a communication system that allows the Commission (as compared to a simple telephone interview) more direct contact with the interested parties and an assessment, even visual, of the environmental context of the subject under assessment. The telematic activity enables the Social Operator to fully acquire the “functioning profile” of the applicant, and the entire Commission can complete the evaluation, also respecting the principle of collegiality, which is a precise regulatory obligation.

Therefore, at present, the described protocol is intended to be applied only to investigations carried out in the following cases:Completion of investigations on documents deemed sufficient for a timely assessment.Completion of home visits in the case of a person who cannot be transported.If a face-to-face medical examination is planned at the local medico-legal units, the interview will take place directly with the interested parties following the visit.It is necessary to emphasize the objectives of the plan as envisaged in the following draft:Replicate the current clinical–organizational processes, so as not to require changes in the existing organization and to be able to coexist with both the activities carried out in person and those provided with telemedicine methods.Keep track of the activities performed, both for the purposes of the legal protection of health personnel and reporting the activities performed.

The strengths of the following action plan are as follows:It does not require the purchase of software and/or hardware with their related costs, time, and commitments both in administrative terms and technological management.Because it is based on widely used software tools, it does not require special training and/or support initiatives for the Institute’s professionals and interested citizens.It allows the Commissions to carry out investigations—where permitted—even in “agile work” mode, with undoubted savings in time and resources, significantly reducing the time related to the transport of the patient and the health personnel, and the economic expenses related to this. Moreover, such a flexible method undoubtedly makes it possible to reduce abstentions and organizational difficulties that would result.

The designed scenario is compatible and can coexist with the procedures and commitments currently in place. Therefore, it can be introduced gradually, with a first pilot phase limited to some subjects referred for assessment (preferably among those most familiar with telematic tools), and then gradually extended, compatible with other priorities and contingencies that may occur in the meantime. As with any informatic project, a gradual approach of this kind is, in any case, also advisable to optimize any refinements that may emerge from daily operations and to record those indicators (commitment, time, satisfaction of the subject under assessment) that will allow the scientific demonstration of the validity and benefits obtainable compared to the procedures in person. Therefore, it is planned to start with the implementation of telematic examinations of those citizens who—directly or through their family members/caregivers—are more familiar with informatic tools.

After the service, the social worker records on a shared Excel document in the office some anonymous indicators about the activity carried out. This allows—both within the Institute and in a broader scenario of collaboration with other health facilities—the evaluation of the effectiveness of the procedure, both from an organizational and health point of view. This phase, although not necessary for the purposes of verification, would be vital, especially in the first period of introduction, as an indirect indicator of the effectiveness of the protocol, as well as useful for the construction of information tools for the future monitoring and updating of procedures [23,24]. Such indicators can help, due to their intrinsic characteristics, administrators and health professionals in highlighting the criticalities of the system to direct priority choices [25,26,27].

Below is a scheme of preliminarily identifiable items, which can be extended in light of specific needs and implemented with aspects of relevance in the specific context (Table 1).

A careful observer might argue that not all the items proposed belong to the sphere of relevance of the social worker’s assessment; however, it should be remembered that to correctly frame and categorize individual cases, it is necessary to collect as much data as possible. These data can possibly be translated into more accurate and specific indicators characterized by high information content and the capability to allow a rapid evaluation of complex phenomena and to guide operational decisions by comparing data over time or space [25].

## 5. Conclusions

Telemedicine has only very recently been introduced into the Italian National Health Service, largely due to the physical limitations induced by the COVID-19 pandemic and new communication technologies, increasingly used in work, social, and, in this case, welfare contexts [28].

In light of the provisions and procedures currently used in the Italian social security sector, the telematic methodology proposed in this paper is an attempt to bring the assessment of the state of disability into a contemporary perspective. This purpose was pursued through the selection of specific contexts that can be easily implemented, while acknowledging that there are still many cases that cannot be visited remotely. The current state already allows us to observe the positive aspects related to the computerization of the process, including the saving of human, professional, temporal, and economic resources, as well as the facilitation of the process even for subjects affected by serious limitations on their autonomy.

The eventual success of the proposed procedures will certainly require careful monitoring, especially in the initial phases, and the collection of numerous data for the construction of performance indicators of the process that will allow constant improvement and updating.

## Figures and Tables

**Table 1 ijerph-21-00253-t001:** Set of monitoring indicators specifically conceived for the first pilot phase.

Field	Specific Indicator
Performance	Physical location of the social security institution.
	Selected Operating Unit/Healthcare structure.
	Healthcare professional (personal data).
	Health discipline specifically investigated.
	Type of benefit, in this case necessarily linked to the assessment of the state of disability.
	Support for the execution of activities by the subject under assessment and the caregiver, if any.
	Date and time of the assessment.
	Duration of service (minutes).
Patient	Age and gender of the patient.
	Municipality of residence of the subject under assessment.
	Type of booking of the visit, based on the applicant.
	Service possibly subject to payment of the ticket.
	It was necessary to interrupt the performance and it was necessary to plan a substitute intervention.
	Complexity of the case (Low, Medium, High).
	Degree of interaction with the subject under investigation.
	Level of satisfaction expressed by the subject under assessment, on a scale of 1 (minimum) to 5 (maximum).
Platform	Device used by the subject under investigation (e.g., smartphone, tablet, personal computer).
	Type of connection of the subject under assessment: Wired–Telephone.
	Communication platform used.
	Audio/video quality level: Low–Medium–High.
	Possible diagnosis code of the subject under investigation.
Other	Other notes.

## Data Availability

The data are not publicly available due to privacy reasons.

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
