# Peer review of "Telehealth Intervention: A Proposal for a Telemedicine Manual to Ascertain the Civil Disability Status in Italy"

_ijerph, 2024, doi:10.3390/ijerph21030253_

Round 1
Reviewer 1 Report
Comments and Suggestions for Authors
The authors present a proposal for a telemedicine manual to determine civil disability status in Italy. The proposal is a telematic method for the evaluation of the disability status by the INPS social worker.
The manuscript requires improving the results to emphasize a scientific discussion.
1. Increase the comparison study of current methods to improve the experience linked to the assessment of the disability status in terms of time and accessibility. Very few references on this topic.
2. In case of use of the manual or protocol in the case of patients who cannot be transferred. How do the authors address the problem of connectivity in distant places? Can the authors ensure full coverage of different platforms such as Skype™?
3. In the conclusions it is necessary to demonstrate: limited cases can optimize the process and times
4. The link: https://www.salute.gov.it/portale/ehealth/dettaglio- ContentEHealth.jsp?lingua=italiano&id=5524&area=eHealth&menu=telemedicina
NOT Available online
5. It is required to demonstrate the entire approval and validation process with patients to verify the expected results: (Available online: time and accessibility
6. There is NO access to the manual to check its contents
7. There is NO access to the interview: questions, results and analysis carried out
8. The so-called “telemedicine” began in the 80s, it is NOT recent
9. What is the construction of process performance indicators?
10. With more and better global references necessary.
Comments on the Quality of English LanguageAuthors must improve English for scientific use.
Check online links.
Author Response
1. Increase the comparison study of current methods to improve the experience linked to the assessment of the disability status in terms of time and accessibility. Very few references on this topic.
Checked
2. In case of use of the manual or protocol in the case of patients who cannot be transferred. How do the authors address the problem of connectivity in distant places? Can the authors ensure full coverage of different platforms such as Skype™?
Further explained
3. In the conclusions it is necessary to demonstrate: limited cases can optimize the process and times
This point has been highlighted inside the manuscript
4. The link: https://www.salute.gov.it/portale/ehealth/dettaglio- ContentEHealth.jsp?lingua=italiano&id=5524&area=eHealth&menu=telemedicina
NOT Available online
Checked
5. It is required to demonstrate the entire approval and validation process with patients to verify the expected results: (Available online: time and accessibility
The present document represent only a proposal
6. There is NO access to the manual to check its contents
This document belongs to INPS, and its content has not been publicly edited yet.
7. There is NO access to the interview: questions, results and analysis carried out
This is not yet the scope of the present document.
8. The so-called “telemedicine” began in the 80s, it is NOT recent
Checked and added a little storiographic focus
9. What is the construction of process performance indicators?
Explained into the manuscript
10. With more and better global references necessary.
Checked
Reviewer 2 Report
Comments and Suggestions for Authors
Thank you for the opportunity to review this manuscript. Presenting and utilising telemedicine and telehealth as a tool to support social workers in assessing environments as part of a disability/handicap evaluation, is very interesting and practical (compared to phonecalls).
However, the manuscript in its current form is not ready for publication.
Major feedback that would need to be rectified before re-submission include:
- There are many sections that need to be re-written with focus on English language.
- There are too few references to support the claims being made throughout the manuscript (particularly introduction).
- A greater understanding of the subject area is needed particularly definitions with citations for telemedicine, and also the claims in conclusion (telemedicine is not new and pre-dates the pandemic).
- The manuscript requires restructuring. For example, the methods section does not clearly state what the methods are, and it is unclear what is being presented as results.
I would encourage the authors to continue their field of research, however the presented manuscript is not ready (at this time) for publication.
Comments on the Quality of English LanguageThe quality of the language is low.
Sentences are very long and in many areas does not make semantic sense.
There are also issues with the content presented in each section not following scientific publication conventions.
Author Response
- There are many sections that need to be re-written with focus on English language.
Checked
- There are too few references to support the claims being made throughout the manuscript (particularly introduction).
Checked
- A greater understanding of the subject area is needed particularly definitions with citations for telemedicine, and also the claims in conclusion (telemedicine is not new and pre-dates the pandemic).
Checked
- The manuscript requires restructuring. For example, the methods section does not clearly state what the methods are, and it is unclear what is being presented as results.
Sections have been implemented
Round 2
Reviewer 1 Report
Comments and Suggestions for Authors
The authors have made all the reviewer's comments and suggestions.
Comments on the Quality of English LanguageSome texts written in the present and others in the past, must unify
Author Response
Thanks to the reviewer for their suggestions. We have made the modifications as requested.

Reviewer 2 Report
Comments and Suggestions for Authors
Thank for your work on this manuscript.
Comments for action:
1. There are too many definitions being used interchangeably in this paper. For example: Telematic, telehealth, telemedicine, telemonitoring, tele sessions, remote monitoring. There are nuances between each term - suggest you unify to one term and define what you mean by that term. If another term is needed to describe something else, describe also what is different about this new term. Message is getting lost amongst terms.
2. There are not enough references to match claims being made in introduction. Specifically:
- Telehealth being used during time of war
- All bullet points following "Telemedicine can contribute to" & "significant potential for"
- History section, sentence "The first examples date back to the nineteenth century"
3. Need to mention if Zoom is GDPR compliant
4. medico-legal Commission and medical-legal commission are used in the document, are these the same or are they different entities?
5. There are a lot of bullet points in this paper. Suggest to tabulate number points on page 9 (Discussion section, following paragraph: Below is a scheme of preliminary items)
6. Conclusion section - it is incorrect to say "so-called", best to remove. Also more clarity needed, perhaps "recent introduction into the Italian National Health Service"
7. English needs improving and introduction of acronyms need to be revisited. For example: INPS acronym not introduced in first instance and then introduced a second time in paper.
Comments on the Quality of English LanguageImproved since last version, but more work is needed.
Author Response

(The authors gave the same response as above.)
